# Microstructures and Mechanical Properties of V-Modified Ti-Zr-Cu-Ni Filler Metals

**DOI:** 10.3390/ma16010199

**Published:** 2022-12-26

**Authors:** Lu Feng, Quanming Liu, Weimin Long, Guoxiang Jia, Haiying Yang, Yangyang Tang

**Affiliations:** 1Xi’an University of Architecture and Technology Huaqing College, Xi’an 710043, China; 2Gas Turbine Technology Department, Xi’an Thermal Power Research Institute Co., Ltd., Xi’an 710054, China; 3State Key Laboratory of Advanced Brazing Filler Metals and Technology, Zhengzhou Research Institute of Mechanical Engineering Co., Ltd., Zhengzhou 450001, China; 4Hanbao Steelmaking Plant, HBIS Group Hansteel Company, Handan 056000, China; 5Institute of Titanium Alloy Research, Northwest Institute for Nonferrous Metal Research, Xi’an 710016, China

**Keywords:** titanium-based filler metal, TA2 titanium alloy, melting properties, the microstructure, tensile properties

## Abstract

TA2 titanium alloy was brazed with Ti-Zr-Cu-Ni-V filler metals developed in a laboratory. The melting properties, the microstructures, phase compositions of filler metals and wettability, erosion properties, tensile properties of the brazed joint were studied in detail. The results show that with the increase of V content, the solidus–liquidus temperature of Ti-Zr-Cu-Ni-V filler metals increased, but the temperature difference basically remained unchanged, trace V element had a limited influence on the melting temperature range of Ti-Zr-Cu-Ni filler metals. The microstructure of Ti-Zr-Cu-Ni-1.5V filler metal was composed of Ti, Zr matrix, (Zr, Cu) solid solution and crystal phase. With the addition of V content, these phases containing V such as Ni_3_VZr_2_, NiV_3_, Ni_2_V in the molten filler metals increased. V was more inclined to combine with Ni to slow down the diffusion of Ni to titanium matrix. The wettability of filler metal with trace (≤0.5 wt.%) V to TA2 titanium alloy became worse, the wettability improved significantly with continuous increase of V content. The thickness of embrittlement layer and intergranular infiltration region decreased significantly by adding V. With the increase of V content, V could regulate the brazing interface reaction, more strengthened phases generated, which resulted the significant increase of the strength (302.72 MPa) and plasticity index (16.3%) of the brazed joint with Ti-Zr-Cu-Ni-1.5V filler metal.

## 1. Introduction

Heat exchanger is a common part of large-scale equipment in aviation, aerospace, navigation and other military fields [1,2,3]. With the increasing demand for lightweight aerospace components and high corrosion resistance of marine materials, titanium exchangers have a wider application prospect in the future [4,5,6,7,8,9,10], which are gradually replacing aluminum alloy, copper alloy and stainless steel exchangers. 

The brazing technology is particularly suitable for the components’ connection with complex shapes, thin walls and combinations of various materials [11,12], which has the advantages of low welding temperature, small deformation of welding parts and high forming accuracy. Titanium-based filler metal is the preferred filler metal to ensure that the properties of titanium weldments [13]. In order to complete brazing work, it is necessary not only to have knowledge of brazing technology, but also to be familiar with all properties of brazed materials themselves [14]. However, so far, there has been a serious shortage of high-quality titanium-based filler metals and corresponding brazing processing for manufacturing titanium exchangers in China [11,15,16,17]. For titanium alloy, maintaining the constitution of the phases in the base and controlling the mechanical properties of the base is important [18]. The long holding time for manufacturing the thin-wall structure is not suggested, and the good ductility of joints of dissimilar materials is a challenge [19]. Above all, the titanium exchanger has a low heat transfer efficiency, temporary life and poor reliability, its popularization and application are limited [20,21]. Related studies of high-quality filler metals and corresponding brazing technology for manufacturing titanium exchangers are very urgent.

The existing Ti-Zr-Cu-Ni alloy system forms a low-melting eutectic composition, which reduced the melting temperature of filler metal, but this filler metal has serious corrosion on the matrix [22,23,24,25,26]. V, Cu and Ni elements have a reasonable proportion to form a low-melting solid solution at room temperature, which slows down the diffusion of Cu and Ni elements to the matrix. In this paper, TA2 titanium alloy was brazed with Ti-Zr-Cu-Ni-V filler metals developed in laboratory. The melting properties, the microstructures, phase compositions of filler metals and wettability, erosion properties, tensile properties of brazed joint were studied in detail. This research will provide technical support towards standardizing filler metals and brazing process for titanium exchangers. The results will also help the precision brazing of titanium tubular, plate fin exchangers, cold plates, waveguides and other military products for aviation, aerospace and navigation.

## 2. Materials and Methods

TA2 titanium alloy was provided by the Northwest Institute for Non-ferrous Metals Research. The beta phase transition temperature of this alloy was measured at 883 °C by the metallographic method. The original size of the TA2 titanium alloy plate was 500 mm × 500 mm × 2 mm (length × width × thickness). The laser cutting method was used to cut the plate longitudinally into several strips with a size of 30 mm × 10 mm × 2 mm and several square blocks with a size of 30 mm × 30 mm × 2 mm, which were used for brazing and wetting tests, respectively. Based on the 37.5Ti-37.5Zr-15Cu-10Ni alloy system, the filler metal was modified by adding appropriate V and precisely optimizing the content of Cu and Ni elements. The design principle of filler metal composition was: (1) keep the content of Ti and Zr main elements unchanged, (2) reduce the content of Cu and Ni elements, increase the content of V elements and ensure that the overall element content was 100%. Filler metal modification was finally completed, the composition optimization is shown in Table 1. Amorphous filler metals are also an important direction of filler metals development in the future [27]. However, the application of titanium-based amorphous filler metals is limited by the cost, size and preparation technology for titanium exchanger. In this paper, the Ti-Zr-Cu-Ni amorphous filler metal developed by laboratory was also used for subsequent TA2-TA2 titanium alloy brazing test. Amorphous filler metal was prepared by single-roller rapidly cooled technology, the width was about 12 mm and the thickness was 45~55 μm, the surface was smooth and clean without color difference.

Titanium-based brazing alloy ingots were prepared by vacuum arc melting, vacuum pressure was kept below 10^−2^ Pa, the ingot was repeatedly smelted 6 times, the ingot with uniform composition was cut by wire-electrode to obtain bulk filler metals, which was used for microstructure analysis and brazing test. The treatment process of the sample to be welded was as follows: (1) Use 400 # and 800 # metallographic sandpaper to polish the welded surface; (2) Place the matrix and bulk filler metal in acetone solution, ultrasonic cleaning for 30 minutes, took out and blew dry for using; (3) Use an electronic scale to weigh 0.05 ± 0.001 g and 0.2 ± 0.002 g of bulk filler metal respectively for brazing and wetting tests, the lap area of brazed joint was 10 mm × 8 mm, the filler metal was fixed around the overlap gap with the binder; (4) Move the assembled sample to be welded as a whole to ZGS-120 high temperature vacuum brazing furnace for heating. As shown in Figure 1, the brazing process was as follows: heating to 300 °C at a heating rate of 15 °C/min, holding for 30 minutes to ensure that the binder was fully volatilized; heating to 750 °C at a heating rate of 10 °C/min, then heating to brazing temperature (900 °C and 930 °C) at a heating rate of 5 °C/min, and holding for 15 minutes; cooling to 300 °C at a cooling rate of 5 °C/min, cooling in furnace to room temperature. 

The solidus–liquidus temperature of filler metal was measured by STA-449F3 TG-DTA/DSC synchronous thermal analyzer (Selb, Germany), 8~29 mg of filler metal was weighed each time to complete the testing under argon protection, and the heating or cooling rate was 15 °C/min, the test temperature range was 25~1000 °C. The phase composition was tested by Rigaku Ultima IV X-ray diffractometer (Wilmington, MA, USA). Phenom XL desktop scanning electron microscope (Eindhoven, The Netherlands) was used to observe the microstructure and morphology of the brazed joint interface, and energy dispersive spectrometer (EDS) equipped with Phenom XL was used to analyze micro-area composition. Wetting tests of new brazing filler metals on TA2 titanium alloy were conducted at 930 °C (brazing temperature)/15 min (holding time) according to GB/T 11364-2008 [28], this test was more than twice under the same conditions. Tensile tests of the brazed joint were carried out according to GB/T 11363-2008 [29], test equipment was MTS GDX300 electronic universal testing machine (Eden Prairie, MN, USA), the strain rate was 0.5 mm/min, the average of three valid test values was taken as the final result of tensile strength and elongation of the brazed joint, the strength error was within 5 MPa, the elongation error was within 0.5 %.

## 3. Results and Discussion

### 3.1. Melting Characteristics of Ti-Zr-Cu-Ni-V Filler Metals

The melting temperature of filler metal is crucial to the selection of brazing temperature. Usually, the brazing temperature is 60~120 °C higher than liquidus temperature of filler metal, while the liquidus temperature is 40~50 °C lower than solidus temperature of the matrix [27]. The brazing temperature should not exceed beta phase transition temperature of titanium matrix. Based on the Ti-Zr alloy system, Cu and Ni elements were introduced to form a low-melting eutectic composition (Ti-Zr-Cu-Ni-V filler metals developed by laboratory), which reduced the melting temperature of the filler metal. The plasticity of filler metal and the structure of interface reaction layer could be improved by reducing the Cu content and adding V element, respectively. 

When the melting temperature is increased, the filler metal will change from solid phase to liquid phase. The solidus temperature (Ts) and liquidus temperature (Tl) of filler metals are the starting point temperature and the ending point temperature of the endothermic peak on the DTA curve, respectively. As shown in Figure 2, all DTA curves of Ti-Zr-Cu-Ni-V filler metals showed only an obvious endothermic peak, filler metal had experienced normal melting and solidification process. Among them, the liquidus temperature of No. 5 filler metal (As shown in Table 1) was the highest, and that of No. 1 filler metal was the lowest. Liquidus temperatures of No. 2. 3. 4 filler metals (adding 0.2, 0.5 and 1.0 wt.% V filler metals) were between the above two. As shown in Table 2, the melting temperature ranges of No. 5 filler metal were from 844.93 °C to 859.05 °C, which was significantly higher than that of No. 1 filler metal. The maximum temperature differences between solidus–liquidus temperatures were 10.39 °C and 13.4 °C, respectively. V element increased the melting temperature of filler metal, but the temperature differences (ΔT) between solidus–liquidus temperatures were basically unchanged, and the influence of trace V on the melting temperature range was very limited. No. 5 filler metal will be used as the brazing filler for subsequent brazing test of TA2-TA2 titanium alloy. As shown in Figure 2, there was no obvious exothermic peak during heating process for No. 6 filler metal (Ti-Zr-Cu-Ni amorphous filler metal), which indicated that there was no complex crystallization process during the melting process. As shown in Table 2, the liquidus temperature of No. 6 filler metal was lower than that of crystalline filler metal, the melting temperature range was only 835.37~840.35 °C, which indicated that the welding process can be completed at a lower temperature and in a shorter time, which is also more favorable for the vacuum brazing of plate-fin titanium alloy exchanger core. 

### 3.2. Microstructure of Ti-Zr-Cu-Ni-V Filler Metals

Figure 3 showed the microstructures of Ti-Zr-Cu-Ni-V crystalline and Ti-Zr-Cu-Ni amorphous filler metals. As shown in Figure 3a,b, the microstructure of Ti-Zr-Cu-Ni-V crystalline filler metal was composed of nearly elliptic bright white phases with grain sizes ranging from 11.9 μm to 32.2 μm. With the increase of V content, the phase composition and grain size of filler metal did not change. As shown in Figure 3c, adding 1.5 wt.% V, the microstructure changed into bright white matrix phase and dark irregular elliptic phase, with the size of dark phase ranging from 3.2 μm to 24.0 μm. V element (≤1.0 wt.%) had a limited effect on the microstructure, which was similar to that of the properties of conventional Ti-Zr-Cu-Ni filler metals. However, adding 1.5 wt.% V, the microstructure of filler metal changed significantly, the effect of V element on the interface reaction of the brazed joint was obviously enhanced. As shown in Figure 3d, there was no obvious grain boundary or grain formation characteristics in the microstructure of No. 6 filler metal, the distribution of each element was relatively uniform, and showed the characteristics of amorphous state.

The microstructure of typical No. 5 filler metal was further analyzed. Figure 4 and Table 3 showed the results of SEM+EDS for filler metal. EDS analysis of point 1 and local area 3 showed that the weight percentage of each element in the matrix was close to the nominal component, the loss or increase of each element was small, and the elements were evenly distributed in the brazing process. EDS analysis of point 2 and local area 4 showed that the weight percentage of elements in the dark gray phase was far from that of the nominal component, and the loss or increase of each element was large. Proportions of Ti and V elements were obviously higher than that of the nominal component, while proportions of Zr, Cu and Ni elements were obviously lower than that of the nominal component. The elements were re-distributed during the brazing melting process. These phases may be solid solution, complex crystal phase and another unknown phase.

### 3.3. Phase Analysis of Ti-Zr-Cu-Ni-V Filler Metals

There are many metallurgical reactions in the smelting process for filler metal containing multiple elements, and resulting in complex phase composition. The phase composition of typical filler metal was analyzed by X-ray diffraction method. As shown in Figure 5a, there was no a very sharp diffraction peak in the XRD diffraction curve of Ti-Zr-Cu-Ni amorphous filler metal, only appeared a diffuse scattering peak with a certain width near 38°, which indicated that amorphous filler metal had no or trace crystalline phase, the alloy had a certain amorphous forming ability. As shown in Figure 5b, the XRD curve of No. 5 filler metal showed several sharp crystal diffraction peaks, the diffraction peaks were wider in the range of 35°~45°. No. 5 filler metal included Ti and Zr matrix phases, (Zr, Cu) solid solution, complex crystal phases and other unknown phases. Crystal phases included Ni_3_VZr_2_, Ni_10_Zr_7_, Cu_10_Zr_7_, CuTi_2_, ZrCu, NiTi_2_, Ti_2_Ni, Zr_2_Ni, CuTi_3_, NiV_3_, Ni_2_V, and so on. The total amount of Cu and Ni elements in No. 5 filler metal was controlled on the basis of No. 1 filler metal. V element increased the phase containing V (Ni_3_VZr_2_, NiV_3_, Ni_2_V) in the molten filler metals. V element tended to combine with Ni element to slow down the diffusion of Ni element into titanium matrix, V element could regulate the brazing interface reaction and improve the strength and toughness of the brazed joint.

### 3.4. The Wettability of Ti-Zr-Cu-Ni-V Filler Metals

The wettability is an important reference to measure welding performance of filler metal. In the process of brazing, the liquid metal filled into the joint gap, which required good wettability to the matrix. According to DTA melting temperature range of Ti-based filler metal, the parameters of 930 °C (brazing temperature)/15 min (holding time) was selected to conduct the wetting test on TA2 titanium alloy. The spreading morphology of Ti-Zr-Cu-Ni-V filler metal was shown in Figure 6. The order of spreading area of five Ti-Zr-Cu-Ni-V filler metals were 0.5 wt.% V < 0.2 wt.% V < 0 wt.% V < 1.0 wt.% V ≈ 1.5 wt.% V. When adding 0.5 and 0.2 wt.% V, filler metals had obvious traces of incomplete spreading, while adding 0, 1.0 and 1.5 wt.% V, filler metal surface was bright and the center had no incomplete spreading trace. The wettability of filler metals with 1.0 and 1.5 wt.% V for TA2 titanium alloy was better, but with 0.5 wt.% V, it was the worst. The wettability of filler metals for TA2 titanium alloy decreased with the addition of trace (≤0.5 wt.%) V, the wettability improved obviously with the continuous increase of V content. With the increase of V content, V element began to form Ni_3_VZr_2_, NiV_3_ and Ni_2_V phases, which regulated the brazing interface reaction, which was conducive to the spreading of filler metal.

### 3.5. Brazing Interface Structure and Erosion Resistance

Erosion of brazing interface is a common problem in brazing titanium [30]. There are two types of interface structures for TA2 titanium alloy brazed joints with Ti-Zr-Cu-Ni-V filler metals: complete reaction type and residual filler metal layer type. Under the same brazing conditions, the formation of complete reaction type should meet the following requirements: less filler metal, narrower joint gap, higher brazing temperature, long holding time, using amorphous foil filler metal, etc. The residual filler metal layer interface was easily formed due to excessive filler metal and relatively wide joint gap. The structure of complete reaction type consisted of element diffusion zone, intergranular infiltration zone and the matrix, the structure of residual filler metal layer added the intermediate reaction zone of filler metal. With the increase of brazing temperature and the extension of holding time, the elements accelerated to penetrate into the matrix, the element diffused at the brazing interface to a certain extent, and formed the brittle layer. However, the elements with high affinity or strong diffusion ability continued to expand to the matrix and form an intergranular infiltration region. 

Embrittlement layer and intergranular infiltration region have important effects on mechanical properties of the brazed joint. The depth of embrittlement layer and the intergranular infiltration distance are important indexes to evaluate the mechanical properties of the brazed joints. Figure 7 showed the interdiffusion distance of interface elements for brazing TA2 titanium alloy by various Ti-based filler metals at 930 °C (brazing temperature)/15 min (holding time). As shown in Figure 7a,b, the thickness of embrittlement layer and intergranular infiltration region of brazed interface with No. 1 and No. 5 filler metals were 59.8 μm, 24.7 μm and 107.9 μm, 52.0 μm, respectively. The thickness of the embrittlement layer and intergranular infiltration region were significantly reduced by adding V element, filler metal with 1.5 wt.% V had a better effect for reducing the erosion of the matrix.

### 3.6. Tensile Properties of the Brazed Joints

The service environments of the weldment have different requirements for filler metal types, which are necessary to study the influence of filler metal components on tensile strength of the brazed joints [31]. Tensile strength is usually used to measure the quality of the brazed joint. Influence of filler metal compositions on the mechanical properties of TA2 titanium alloy brazed joint was studied. No. 1., No. 5 and No. 6 filler metals were used to braze TA2 titanium alloy at 930 °C (brazing temperature)/15 min (holding time), respectively. Figure 8 showed the relationship between tensile strength and filler metal components. Table 4 showed tensile strength and elongation of the brazed joint with different brazing filler component. Tensile strengths of the above three brazed joints were 294.47, 302.72 and 287.31 MPa, respectively. Compared with No. 1 filler metal, the strength of the brazed joint with No. 5 filler metal was increased by 8.25 MPa. However, using No. 6 filler metal, the strength of the brazed joint decreased by 7.16 MPa. The elongation data indicate that the order of plasticity index was No. 5 > No. 1 > No. 6 filler metal. With the increase of V content, V could regulate the brazing interface reaction, more strengthened phases generated, which resulted in the strength of the brazed joint greatly increased. No. 5 filler metal had good tensile strength (302.72 MPa) and plasticity index (16.3%), which was an ideal filler metal for brazing TA2 titanium alloy. 

## 4. Conclusions

(1) V element increased the melting temperature of filler metals, but the difference between solidus–liquidus temperatures was basically unchanged, the influence of trace V on melting temperature range was limited.

(2) The addition of (≤1.0 wt.%) V element had a limited effect on the microstructure, adding 1.5 wt.% V, the microstructure changed into bright white matrix phase and dark irregular elliptic phase, with the size of dark phase ranging from 3.2 μm to 24.0 μm.

(3) The microstructure of Ti-Zr-Cu-Ni-1.5V filler metal included Ti and Zr matrix phases, (Zr, Cu) solid solution, complex crystal phases. V element tended to combine with Ni element to slow down the diffusion of Ni element into titanium matrix and improved the strength and toughness of the brazed joint.

(4) The wettability of filler metals with 1.0 and 1.5 wt.% V for TA2 titanium alloy were better, but with 0.5 wt.% V was the worst. With the addition of trace (≤0.5 wt.%) V, the wettability decreased gradually, the wettability improved obviously with continuous increase of V content.

(5) Adding V content, the thickness of embrittlement layer and intergranular infiltration region were significantly reduced, 1.5 wt.% V had a better effect for reducing the erosion of the matrix.

(6) With the increase of V content, V could regulate the brazing interface reaction, more strengthened phases generated, which resulted the significant increase of the strength (302.72 MPa) and plasticity index (16.3%) of brazed joint with Ti-Zr-Cu-Ni-1.5V filler metal. 

## Figures and Tables

**Figure 1 materials-16-00199-f001:**
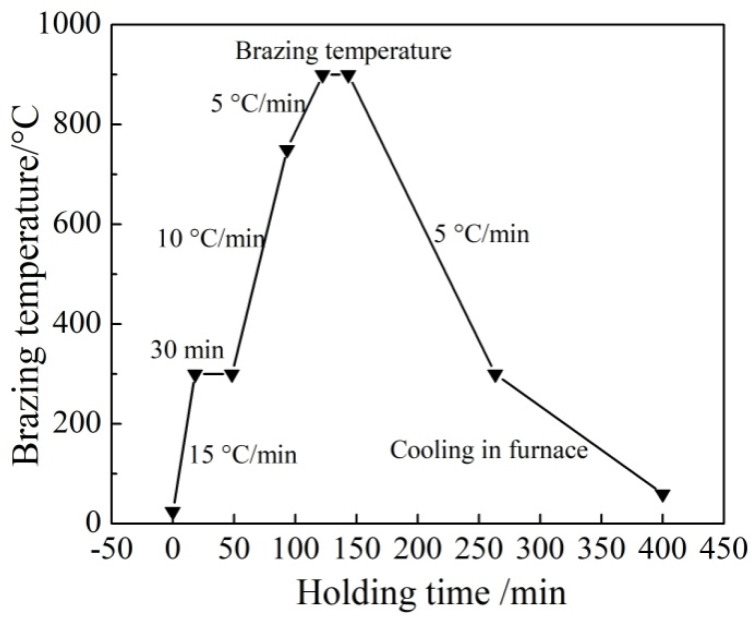
Flow chart of brazing process.

**Figure 2 materials-16-00199-f002:**
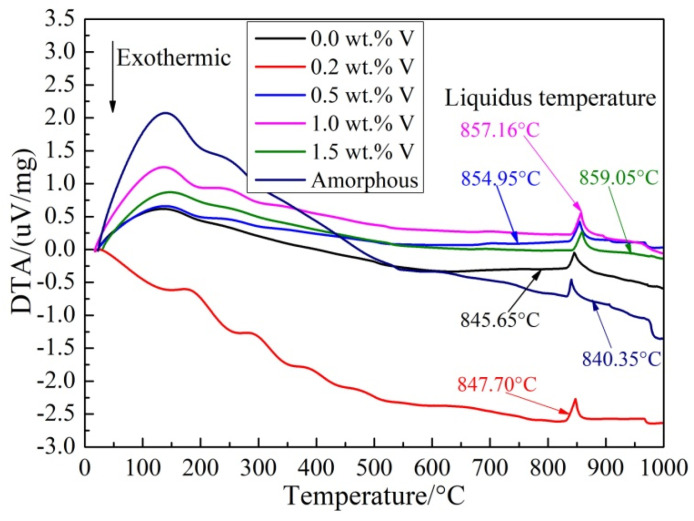
DTA curves of Ti-Zr-Cu-Ni-V filler metals.

**Figure 3 materials-16-00199-f003:**
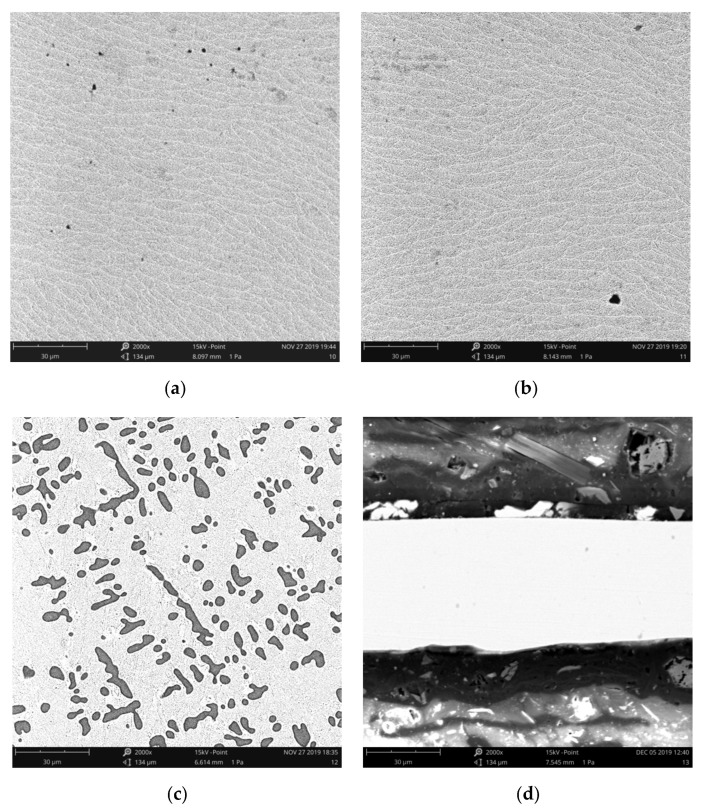
Microstructures of Ti-Zr-Cu-Ni-V crystalline and Ti-Zr-Cu-Ni amorphous filler metals: (**a**) No. 1, (**b**) No. 4, (**c**) No. 5, (**d**) No. 6.

**Figure 4 materials-16-00199-f004:**
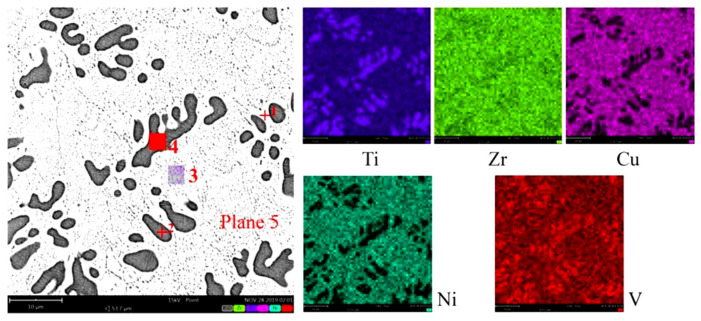
Element distribution of No. 5 filler metal.

**Figure 5 materials-16-00199-f005:**
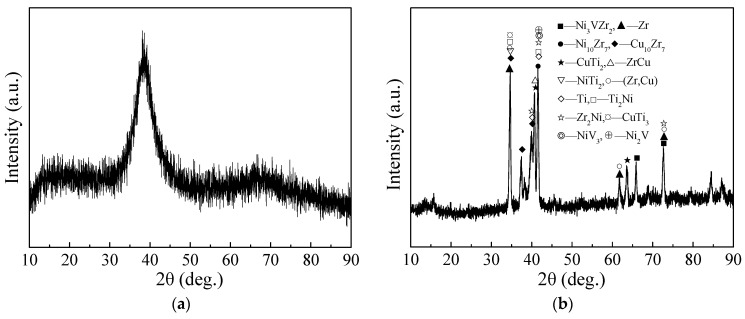
XRD diffraction curves of filler metals: (**a**) No. 6, (**b**) No. 5.

**Figure 6 materials-16-00199-f006:**
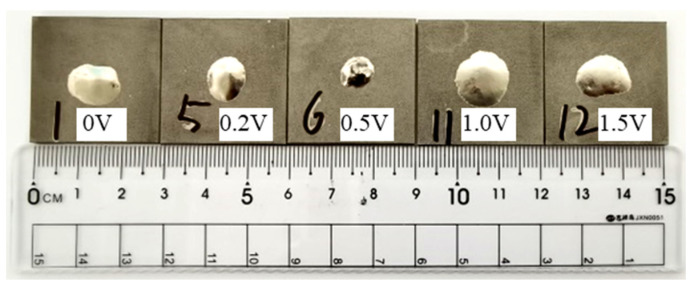
Spreading morphology of Ti-Zr-Cu-Ni-V filler metals for TA2 titanium alloy.

**Figure 7 materials-16-00199-f007:**
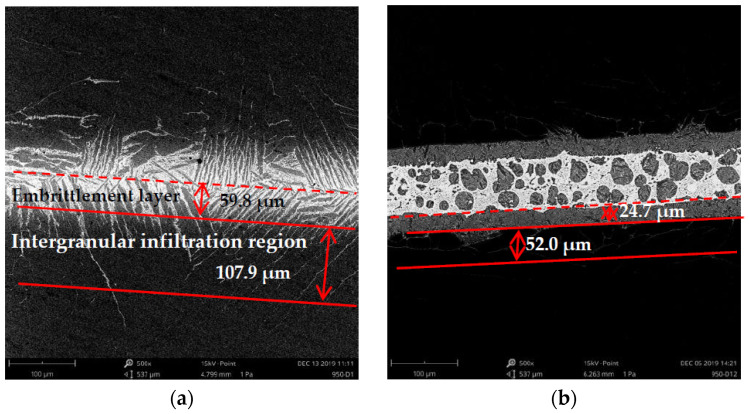
Inter-diffusion distance of elements at brazing interface: (**a**) No. 1, (**b**) No. 5.

**Figure 8 materials-16-00199-f008:**
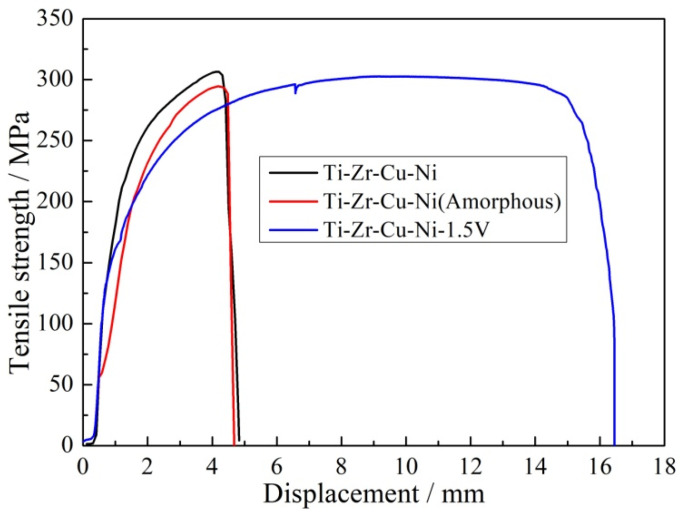
Relationship between tensile strength and filler metal components.

**Table 1 materials-16-00199-t001:** Composition optimization of new titanium-based filler metals.

Different Fillers No.	Ti (wt.%)	Zr (wt.%)	Cu (wt.%)	Ni (wt.%)	V (wt.%)
1	37.5	37.5	15	10	0
2	37.5	37.5	14.9	9.9	0.2
3	37.5	37.5	14.75	9.75	0.5
4	37.5	37.5	14.5	9.5	1.0
5	37.5	37.5	14.25	9.25	1.5

**Table 2 materials-16-00199-t002:** Melting temperature ranges of new titanium-based filler metals.

Different Fillers No.	Main Components wt.%	V wt.%	Ts °C	Tl °C	ΔT °C
1	Ti-37.5Zr-15Cu-10Ni	0	834.65	845.65	11
2	Ti-37.5Zr-14.9Cu-9.9Ni	0.2	834.54	847.70	13.16
3	Ti-37.5Zr-14.75Cu-9.75Ni	0.5	841.25	854.95	13.7
4	Ti-37.5Zr-14.5Cu-9.5Ni	1.0	842.70	857.16	14.46
5	Ti-37.5Zr-14.25Cu-9.25Ni	1.5	844.93	859.05	14.12
6	Ti-37.5Zr-15Cu-10Ni (Amorphous)	0	835.37	840.35	4.98

Note: solidus temperature (Ts), liquidus temperature (Tl), temperature difference between solidus-liquidus temperatures (ΔT).

**Table 3 materials-16-00199-t003:** Element content of each phase in No. 5 filler metals.

Site No.	Weight Conc. (wt.%)				
	Ti	Zr	Cu	Ni	V
1 (Point)	30.30	38.95	17.53	11.70	1.52
2 (Point)	59.62	30.56	5.66	1.93	2.22
3 (Local)	34.09	38.75	15.75	10.10	1.30
4 (Local)	60.66	29.99	5.31	1.58	2.45
5 (Plane)	38.55	37.28	14.10	8.68	1.38

**Table 4 materials-16-00199-t004:** Tensile strength and elongation of the brazed joints under different composition.

Different Fillers	Tensile Strength /MPa	Elongation /%
Ti-Zr-Cu-Ni	294.47	8.8
Ti-Zr-Cu-Ni-1.5V	302.72	16.3
Ti-Zr-Cu-Ni (Amorphous)	287.31	4.7

Note: The average of three valid test values was taken as the final result of tensile strength and elongation of the brazed joint, strength error is within 5 MPa, elongation error is within 0.5%.

## Data Availability

The data presented in this study are available on request from the corresponding author.

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
