# Peer review of "Microstructures and Mechanical Properties of V-Modified Ti-Zr-Cu-Ni Filler Metals"

_materials, 2022, doi:10.3390/ma16010199_

Round 1

Reviewer 1 Report

The authors have presented a systematic study. Experimental work is good. I recommend a thorough review of english language throughout the manuscript.  The following suggestions and queries are to be addressed to enhance the readability and quality of this manuscript.

Abstract:

1.      Line 17 and 18: repetition of ‘and’ makes the sentence looks odd. Please check and correct accordingly.

2.    Line 29: ‘which resulted ……………………..increased’ : check for English language and missing words.

Introduction:

3.  The English language requires a thorough proof reading and overall improvement in the entire manuscript.

4.    The introduction is mostly written in past tense of English language which makes it difficult to read and understand. Please check as the current trend cannot be written in the past tense. e.g. Line 46.

5.  Can the word homemade be replaced with laboratory developed or something else which sounds more technical than homemade?

6. Line 54: ‘This research…………………….and navigation’, this sentence can be divided into 2 sentences.

Materials & Methods

7. How was the composition of the new titanium based filler metal (Table 1) determined? Which methodology was adopted. Mention this in the manuscript.

8.  Line 88: Give the brazing process in a tabular form for better readability.

9.  For STA, the cooling rate is given in K/min whereas the start and end temp is in degree Celsius. Can both the units be made same for better readability?

10.  Tensile rate??? Is this the correct technical term? or should it be the strain rate?

Results & Discussion

111.   Line 114: Can the word ‘self made’ be replaced with ‘laboratory developed’ or something technical. This must be uniform throughout.

12.   Figure 1: What is the reason behind a distinctive and different trend curve for 0.2wt.% V? This is the only one that appears to be going all negative with no distinct peak in the initial region. Please explain.

13.   Below Table 2 , mention the extended form of abbreviation used in the Table.

14.   Line 170: What is the meaning of obvious grain characteristic? Explain properly in the text.

15.   Figure 4: The noise in XRD is significantly large? Can you reduce the noise?

16.   How many tensile specimens were tested for each type? One sample of each type is not enough. In Table 4 have you reported the mean value? Mention the standard deviation too.

Author Response

Dear Reviewer,

Please find the response in the attachment

Reviewer 2 Report

This paper examines the microstructure and mechanical properties of hand-made fillers that the authors of the article have developed for titanium alloys. Overall, the article is well-written and the reported results are of great scientific and practical importance. However, before it is recommended to be published, it needs to be improved, especially in terms of the English language.

I have attached my comments in a separate PDF file.

Author Response

Dear Reviewer,

Please find the response file in the attachment.

Round 2

Reviewer 2 Report

The authors addressed all questions extensively. I recommend the article for publication in its present form.